# Where Do *You* Think You're Going?: Inferring Beliefs about Dynamics from Behavior

**Siddharth Reddy, Anca D. Dragan, Sergey Levine**
Department of Electrical Engineering and Computer Science
University of California, Berkeley
{sgr,anca,svlevine}@berkeley.edu

## Abstract

Inferring intent from observed behavior has been studied extensively within the frameworks of Bayesian inverse planning and inverse reinforcement learning. These methods infer a goal or reward function that best explains the actions of the observed agent, typically a human demonstrator. Another agent can use this inferred intent to predict, imitate, or assist the human user. However, a central assumption in inverse reinforcement learning is that the demonstrator is close to optimal. While models of suboptimal behavior exist, they typically assume that suboptimal actions are the result of some type of random noise or a known cognitive bias, like temporal inconsistency. In this paper, we take an alternative approach, and model suboptimal behavior as the result of internal model misspecification: the reason that user actions might deviate from near-optimal actions is that the user has an incorrect set of beliefs about the rules – the dynamics – governing how actions affect the environment. Our insight is that while demonstrated actions may be suboptimal in the real world, they may actually be near-optimal with respect to the user's *internal* model of the dynamics. By estimating these internal beliefs from observed behavior, we arrive at a new method for inferring intent. We demonstrate in simulation and in a user study with 12 participants that this approach enables us to more accurately model human intent, and can be used in a variety of applications, including offering assistance in a shared autonomy framework and inferring human preferences.

## 1 Introduction

Characterizing the drive behind human actions in the form of a goal or reward function is broadly useful for predicting future behavior, imitating human actions in new situations, and augmenting human control with automated assistance – critical functions in a wide variety of applications, including pedestrian motion prediction [57], virtual character animation [38], and robotic teleoperation [35]. For example, remotely operating a robotic arm to grasp objects can be challenging for a human user due to unfamiliar or unintuitive dynamics of the physical system and control interface. Existing frameworks for assistive teleoperation and shared autonomy aim to help users perform such tasks [35, 29, 46, 8, 45]. These frameworks typically rely on existing methods for intent inference in the sequential decision-making context, which use Bayesian inverse planning or inverse reinforcement learning to learn the user's goal or reward function from observed control demonstrations. These methods typically assume that user actions are near-optimal, and deviate from optimality due to random noise [56], specific cognitive biases in planning [16, 15, 4], or risk sensitivity [33].

---

See https://sites.google.com/view/inferring-internal-dynamics for supplementary materials, including videos and code.

The key insight in this paper is that suboptimal behavior can also arise from a mismatch between the dynamics of the real world and the user's internal beliefs of the dynamics, and that a user policy that appears suboptimal in the real world may actually be near-optimal with respect to the user's internal dynamics model. As resource-bounded agents living in an environment of dazzling complexity, humans rely on intuitive theories of the world to guide reasoning and planning [21, 26]. Humans leverage internal models of the world for motor control [53, 30, 14, 34, 49], goal-directed decision making [7], and representing the mental states of other agents [39]. Simplified internal models can systematically deviate from the real world, leading to suboptimal behaviors that have unintended consequences, like hitting a tennis ball into the net or skidding on an icy road. For example, a classic study in cognitive science shows that human judgments about the physics of projectile motion are closer to Aristotelian impetus theory than to true Newtonian dynamics – in other words, people tend to ignore or underestimate the effects of inertia [11]. Characterizing the gap between internal models and reality by modeling a user's internal predictions of the effects of their actions allows us to better explain observed user actions and infer their intent.

The main contribution of this paper is a new algorithm for intent inference that first estimates a user's internal beliefs of the dynamics of the world using observations of how they act to perform known tasks, then leverages the learned internal dynamics model to infer intent on unknown tasks. In contrast to the closest prior work [28, 22], our method scales to problems with high-dimensional, continuous state spaces and nonlinear dynamics. Our internal dynamics model estimation algorithm assumes the user takes actions with probability proportional to their exponentiated soft Q-values. We fit the parameters of the internal dynamics model to maximize the likelihood of observed user actions on a set of tasks with known reward functions, by tying the internal dynamics to the soft Q function via the soft Bellman equation. At test time, we use the learned internal dynamics model to predict the user's desired next state given their current state and action input.

We run experiments first with simulated users, testing that we can recover the internal dynamics, even in MDPs with a continuous state space that would otherwise be intractable for prior methods. We then run a user study with 12 participants in which humans play the Lunar Lander game (screenshot in Figure 1). We recover a dynamics model that explains user actions better than the real dynamics, which in turn enables us to assist users in playing the game by transferring their control policy from the recovered internal dynamics to the real dynamics.

## 2  Background

Inferring intent in sequential decision-making problems has been heavily studied under the framework of inverse reinforcement learning (IRL), which we build on in this work. The aim of IRL is to learn a user's reward function from observed control demonstrations. IRL algorithms are not directly applicable to our problem of learning a user's beliefs about the dynamics of the environment, but they provide a helpful starting point for thinking about how to extract hidden properties of a user from observations of how they behave.

In our work, we build on the maximum causal entropy (MaxCausalEnt) IRL framework [55, 6, 44, 36, 28]. In an MDP with a discrete action space $\mathcal{A}$, the human demonstrator is assumed to follow a policy $\pi$ that maximizes an entropy-regularized reward $R(s, a, s')$ under dynamics $T(s'|s, a)$. Equivalently,

$$\pi(a|s) \triangleq \frac{\exp\left(Q(s, a)\right)}{\sum_{a' \in \mathcal{A}} \exp\left(Q(s, a')\right)}, \tag{1}$$

where $Q$ is the soft $Q$ function, which satisfies the soft Bellman equation [55],

$$Q(s, a) = \mathbb{E}_{s' \sim T(\cdot|s,a)}\left[R(s, a, s') + \gamma V(s')\right], \tag{2}$$

with $V$ the soft value function,

$$V(s) \triangleq \log\left(\sum_{a \in \mathcal{A}} \exp\left(Q(s, a)\right)\right). \tag{3}$$

Prior work assumes $T$ is the true dynamics of the real world, and fits a model of the reward $R$ that maximizes the likelihood (given by Equation 1) of some observed demonstrations of the user acting in the real world. In our work, we assume access to a set of training tasks for which the rewards $R$ are known, fit a model of the internal dynamics $T$ that is allowed to deviate from the real dynamics, then use the recovered dynamics to infer intent (e.g., rewards) in new tasks.

# 3   Internal Dynamics Model Estimation

We split up the problem of intent inference into two parts: learning the internal dynamics model from user demonstrations on known tasks (the topic of this section), and using the learned internal model to infer intent on unknown tasks (discussed later in Section 4). We assume that the user's internal dynamics model is stationary, which is reasonable for problems like robotic teleoperation when the user has some experience practicing with the system but still finds it unintuitive or difficult to control. We also assume that the real dynamics are known ex-ante or learned separately.

Our aim is to recover a user's implicit beliefs about the dynamics of the world from observations of how they act to perform a set of tasks. The key idea is that, when their internal dynamics model deviates from the real dynamics, we can no longer simply fit a dynamics model to observed state transitions. Standard dynamics learning algorithms typically assume access to $(s, a, s')$ examples, with $(s, a)$ features and $s'$ labels, that can be used to train a classification or regression model $p(s'|s, a)$ using supervised learning. In our setting, we instead have $(s, a)$ pairs that indirectly encode the state transitions that the user expected to happen, but did not necessarily occur, because the user's internal model predicted different outcomes $s'$ than those that actually occurred in the real world. Our core assumption is that the user's policy is near-optimal with respect to the unknown internal dynamics model. To this end, we propose a new algorithm for learning the internal dynamics from action demonstrations: inverse soft Q-learning.

## 3.1   Inverse Soft Q-Learning

The key idea behind our algorithm is that we can fit a parametric model of the internal dynamics model $T$ that maximizes the likelihood of observed action demonstrations on a set of training tasks with known rewards by using the soft $Q$ function as an intermediary.[1] We tie the internal dynamics $T$ to the soft $Q$ function via the soft Bellman equation (Equation 2), which ensures that the soft $Q$ function is induced by the internal dynamics $T$. We tie the soft $Q$ function to action likelihoods using Equation 1, which encourages the soft $Q$ function to explain observed actions. We accomplish this by solving a constrained optimization problem in which the demonstration likelihoods appear in the objective and the soft Bellman equation appears in the constraints.

**Formulating the optimization problem.**   Assume the action space $\mathcal{A}$ is discrete.[2]   Let $i \in \{1, 2, ..., n\}$ denote the training task, $R_i(s, a, s')$ denote the known reward function for task $i$, $T$ denote the unknown internal dynamics, and $Q_i$ denote the unknown soft $Q$ function for task $i$. We represent $Q_i$ using a function approximator $Q_{\boldsymbol{\theta}_i}$ with parameters $\boldsymbol{\theta}_i$, and the internal dynamics using a function approximator $T_{\boldsymbol{\phi}}$ parameterized by $\boldsymbol{\phi}$. Note that, while each task merits a separate soft $Q$ function since each task has different rewards, all tasks share the same internal dynamics.

Recall the soft Bellman equation (Equation 2), which constrains $Q_i$ to be the soft $Q$ function for rewards $R_i$ and internal dynamics $T$. An equivalent way to express this condition is that $Q_i$ satisfies $\delta_i(s, a) = 0 \ \forall s, a$, where $\delta_i$ is the soft Bellman error:

$$\delta_i(s, a) \triangleq Q_i(s, a) - \int_{s' \in \mathcal{S}} T(s'|s, a) \left( R_i(s, a, s') + \gamma V_i(s') \right) \mathrm{d}s'. \tag{4}$$

We impose the same condition on $Q_{\boldsymbol{\theta}_i}$ and $T_{\boldsymbol{\phi}}$, i.e., $\delta_{\boldsymbol{\theta}_i, \boldsymbol{\phi}}(s, a) = 0 \ \forall s, a$. We assume our representations are expressive enough that there exist values of $\boldsymbol{\theta}_i$ and $\boldsymbol{\phi}$ that satisfy the condition. We fit parameters $\boldsymbol{\theta}_i$ and $\boldsymbol{\phi}$ to maximize the likelihood of the observed demonstrations while respecting the soft Bellman equation by solving the constrained optimization problem

$$\underset{\{\boldsymbol{\theta}_i\}_{i=1}^n, \boldsymbol{\phi}}{\text{minimize}} \ \sum_{i=1}^n \sum_{(s,a) \in \mathcal{D}_i^{\text{demo}}} - \log \pi_{\boldsymbol{\theta}_i}(a|s) \tag{5}$$

$$\text{subject to} \ \delta_{\boldsymbol{\theta}_i, \boldsymbol{\phi}}(s, a) = 0 \ \forall i \in \{1, 2, ..., n\}, s \in \mathcal{S}, a \in \mathcal{A},$$

where $\mathcal{D}_i^{\text{demo}}$ are the demonstrations for task $i$, and $\pi_{\boldsymbol{\theta}_i}$ denotes the action likelihood given by $Q_{\boldsymbol{\theta}_i}$ and Equation 1.

**Solving the optimization problem.** We use the penalty method [5] to approximately solve the constrained optimization problem described in Equation 5, which recasts the problem as unconstrained optimization of the cost function

$$c(\boldsymbol{\theta}, \boldsymbol{\phi}) \triangleq \sum_{i=1}^{n} \sum_{(s,a) \in \mathcal{D}_i^{\text{demo}}} -\log \pi_{\boldsymbol{\theta}_i}(a|s) + \frac{\rho}{2} \sum_{i=1}^{n} \int_{s \in \mathcal{S}} \sum_{a \in \mathcal{A}} (\delta_{\boldsymbol{\theta}_i, \boldsymbol{\phi}}(s,a))^2 \mathrm{d}s, \qquad (6)$$

where $\rho$ is a constant hyperparameter, $\pi_{\boldsymbol{\theta}_i}$ denotes the action likelihood given by $Q_{\boldsymbol{\theta}_i}$ and Equation 1, and $\delta_{\boldsymbol{\theta}_i, \boldsymbol{\phi}}$ denotes the soft Bellman error, which relates $Q_{\boldsymbol{\theta}_i}$ to $T_{\boldsymbol{\phi}}$ through Equation 4.

For MDPs with a discrete state space $\mathcal{S}$, we minimize the cost as is. MDPs with a continuous state space present two challenges: (1) an intractable integral over states in the sum over penalty terms, and (2) integrals over states in the expectation terms of the soft Bellman errors $\delta$ (recall Equation 4). To tackle (1), we resort to constraint sampling [10]; specifically, randomly sampling a subset of state-action pairs $\mathcal{D}_i^{\text{samp}}$ from rollouts of a random policy in the real world. To tackle (2), we choose a deterministic model of the internal dynamics $T_{\boldsymbol{\phi}}$, which simplifies the integral over next states in Equation 4 to a single term[3].

In our experiments, we minimize the objective in Equation 6 using Adam [31]. We use a mix of tabular representations, structured linear models, and relatively shallow multi-layer perceptrons to model $Q_{\boldsymbol{\theta}_i}$ and $T_{\boldsymbol{\phi}}$. In the tabular setting, $\boldsymbol{\theta}_i$ is a table of numbers with a separate entry for each state-action pair, and $\boldsymbol{\phi}$ can be a table with an entry between 0 and 1 for each state-action-state triple. For linear and neural network representations, $\boldsymbol{\theta}_i$ and $\boldsymbol{\phi}$ are sets of weights.

### 3.2 Regularizing the Internal Dynamics Model

One issue with our approach to estimating the internal dynamics is that there tend to be multiple feasible internal dynamics models that explain the demonstration data equally well, which makes the correct internal dynamics model difficult to identify. We propose two different solutions to this problem: collecting demonstrations on multiple training tasks, and imposing a prior on the learned internal dynamics that encourages it to be similar to the real dynamics.

**Multiple training tasks.** If we only collect demonstrations on $n = 1$ training tasks, then at any given state $s$ and action $a$, the recovered internal dynamics may simply assign a likelihood of one to the next state $s'$ that maximizes the reward function $R_1(s, a, s')$ of the single training task. Intuitively, if our algorithm is given user demonstrations on only one task, then the user's actions can be explained by an internal dynamics model that always predicts the best possible next state for that one task (e.g., the target in a navigation task), no matter the current state or user action. We can mitigate this problem by collecting demonstrations on $n > 1$ training tasks, which prevents degenerate solutions by forcing the internal dynamics to be consistent with a diverse set of user policies.

**Action intent prior.** In our experiments, we also explore another way to regularize the learned internal dynamics: imposing the prior that the learned internal dynamics $T_{\boldsymbol{\phi}}$ should be similar to the known real dynamics $T^{\text{real}}$ by restricting the support of $T_{\boldsymbol{\phi}}(\cdot|s, a)$ to states $s'$ that are reachable in the real dynamics. Formally,

$$T_{\boldsymbol{\phi}}(s'|s, a) \triangleq \sum_{a^{\text{int}} \in \mathcal{A}} T^{\text{real}}(s'|s, a^{\text{int}}) f_{\boldsymbol{\phi}}(a^{\text{int}}|s, a) \qquad (7)$$

where $a$ is the user's action, $a^{\text{int}}$ is the user's intended action, and $f_{\boldsymbol{\phi}} : \mathcal{S} \times \mathcal{A}^2 \to [0, 1]$ captures the user's 'action intent' – the action they would have taken if they had perfect knowledge of the real dynamics. This prior changes the structure of our internal dynamics model to predict the user's intended action with respect to the real dynamics, rather than directly predicting their intended next state. Note that, when we use this action intent prior, $T_{\boldsymbol{\phi}}$ is no longer directly modeled. Instead, we model $f_{\boldsymbol{\phi}}$ and use Equation 7 to compute $T_{\boldsymbol{\phi}}$.

In our experiments, we examine the effects of employing multiple training tasks and imposing the action intent prior, together and in isolation.

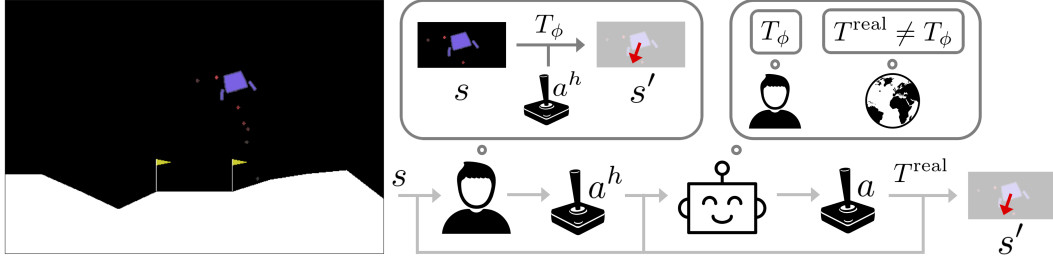

**Figure 1:** A high-level schematic of our internal-to-real dynamics transfer algorithm for shared autonomy, which uses the internal dynamics model learned by our method to assist the user with an unknown control task; in this case, landing the lunar lander between the flags. The user's actions are assumed to be consistent with their internal beliefs about the dynamics $T_\phi$, which differ from the real dynamics $T^{\text{real}}$. Our system models the internal dynamics to determine where the user is trying to go next, then acts to get there.

## 4    Using Learned Internal Dynamics Models

The ability to learn internal dynamics models from demonstrations is broadly useful for intent inference. In our experiments, we explore two applications: (1) shared autonomy, in which a human and robot collaborate to solve a challenging real-time control task, and (2) learning the reward function of a user who generates suboptimal demonstrations due to internal model misspecification. In (1), intent is formalized as the user's desired next state, while in (2), the user's intent is represented by their reward function.

### 4.1    Shared Autonomy via Internal-to-Real Dynamics Transfer

Many control problems involving human users are challenging for autonomous agents due to partial observability and imprecise task specifications, and are also challenging for humans due to constraints such as bounded rationality [48] and physical reaction time. Shared autonomy combines human and machine intelligence to perform control tasks that neither can on their own, but existing methods have the basic requirement that the machine either needs a description of the task or feedback from the user, e.g., in the form of rewards [29, 8, 45]. We propose an alternative algorithm that assists the user without knowing their reward function by leveraging the internal dynamics model learned by our method. The key idea is formalizing the user's intent as their desired next state. We use the learned internal dynamics model to infer the user's desired next state given their current state and control input, then execute an action that will take the user to the desired state under the real dynamics; essentially, transferring the user's policy from the internal dynamics to the real dynamics, akin to simulation-to-real transfer for robotic control [13]. See Figure 1 for a high-level schematic of this process.

Equipped with the learned internal dynamics model $T_\phi$, we perform internal-to-real dynamics transfer by observing the user's action input, computing the induced distribution over next states using the internal dynamics, and executing an action that induces a similar distribution over next states in the real dynamics. Formally, for user control input $a_t^h$ and state $s_t$, we execute action $a_t$, where

$$a_t \triangleq \underset{a \in \mathcal{A}}{\arg\min}\, D_{\text{KL}}(T_\phi(s_{t+1}|s_t, a_t^h) \,\|\, T^{\text{real}}(s_{t+1}|s_t, a)) \tag{8}$$

### 4.2    Learning Rewards from Misguided User Demonstrations

Most existing inverse reinforcement learning algorithms assume that the user's internal dynamics are equivalent to the real dynamics, and learn their reward function from near-optimal demonstrations. We explore a more realistic setting in which the user's demonstrations are suboptimal due to a mismatch between their internal dynamics and the real dynamics. Users are 'misguided' in that their behavior is suboptimal in the real world, but near-optimal with respect to their internal dynamics. In this setting, standard IRL algorithms that do not distinguish between the internal and the real dynamics learn incorrect reward functions. Our method can be used to learn the internal dynamics, then explicitly incorporate the learned internal dynamics into an IRL algorithm's behavioral model of the user.

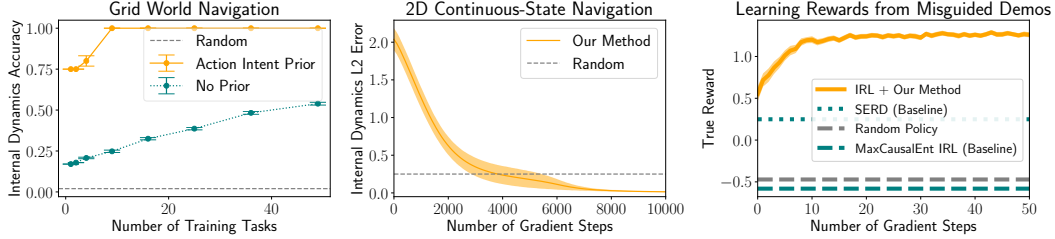

**Figure 2: Left, Center:** Error bars show standard error on ten random seeds. Our method learns accurate internal dynamics models, the regularization methods in Section 3.2 increase accuracy, and the approximations for continuous-state MDPs in Section 3.1 do not compromise accuracy. **Right:** Error regions show standard error on ten random tasks and ten random seeds each. Our method learns an internal dynamics model that enables MaxCausalEnt IRL to learn rewards from misguided user demonstrations.

In our experiments, we instantiate prior work with MaxCausalEnt IRL [55], which inverts the behavioral model from Equation 1 to infer rewards from demonstrations. We adapt it to our setting, in which the real dynamics are known and the internal dynamics are either learned (separately by our algorithm) or assumed to be the same as the known real dynamics. MaxCausalEnt IRL cannot learn the user's reward function from misguided demonstrations when it makes the standard assumption that the internal dynamics are equal to the real dynamics, but can learn accurate rewards when it instead uses the learned internal dynamics model produced by our algorithm.

# 5    User Study and Simulation Experiments

The purpose of our experiments is two-fold: (1) to test the correctness of our algorithm, and (2) to test our core assumption that a human user's internal dynamics can be different from the real dynamics, and that our algorithm can learn an internal dynamics model that is useful for assisting the user through internal-to-real dynamics transfer. To accomplish (1), we perform three simulated experiments that apply our method to shared autonomy (see Section 4.1) and to learning rewards from misguided user demonstrations (see Section 4.2). In the shared autonomy experiments, we first use a tabular grid world navigation task to sanity-check our algorithm and analyze the effects of different regularization choices from Section 3.2. We then use a continuous-state 2D navigation task to test our method's ability to handle continuous observations using the approximations described in Section 3.1. In the reward learning experiment, we use the grid world environment to compare the performance of MaxCausalEnt IRL [55] when it assumes the internal dynamics are the same as the real dynamics to when it uses the internal dynamics learned by our algorithm. To accomplish (2), we conduct a user study in which 12 participants play the Lunar Lander game (see Figure 1) with and without internal-to-real dynamics transfer assistance. We summarize these experiments in Sections 5.1 and 5.2. Further details are provided in Section 9.1 of the appendix.

## 5.1    Simulation Experiments

**Shared autonomy.** The grid world provides us with a domain where exact solutions are tractable, which enables us to verify the correctness of our method and compare the quality of the approximation in Section 3.1 with an exact solution to the learning problem. The continuous task provides a more challenging domain where exact solutions via dynamic programming are intractable. In each setting, we simulate a user with an internal dynamics model that is severely biased away from the real dynamics of the simulated environment. The simulated user's policy is near-optimal with respect to their internal dynamics, but suboptimal with respect to the real dynamics. Figure 2 (left and center plots) provides overall support for the hypothesis that our method can effectively learn tabular and continuous representations of the internal dynamics for MDPs with discrete and continuous state spaces. The learned internal dynamics models are accurate with respect to the ground truth internal dynamics, and internal-to-real dynamics transfer successfully assists the simulated users. The learned internal dynamics model becomes more accurate as we increase the number of training tasks, and the action intent prior (see Section 3.2) increases accuracy when the internal dynamics are similar to the real dynamics. These results confirm that our approximate algorithm is correct and yields solutions

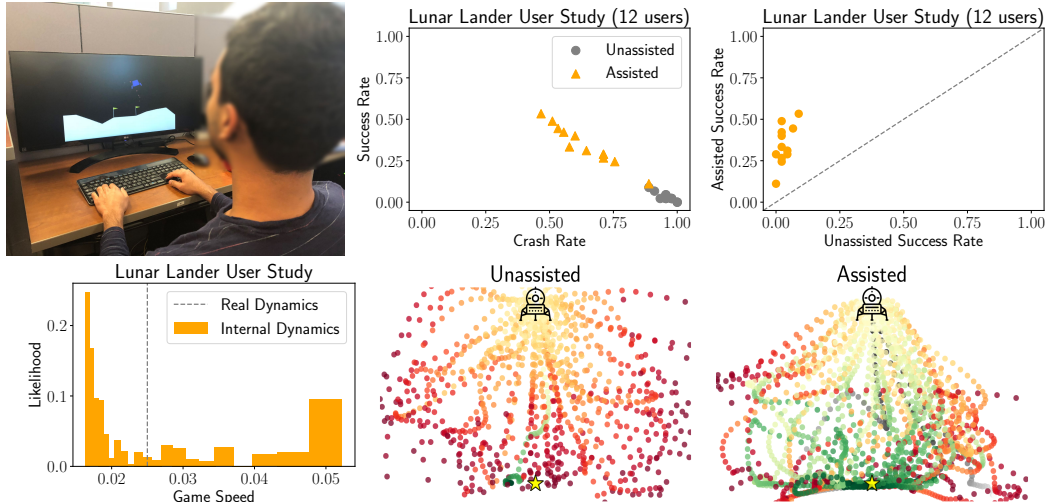

**Figure 3:** Human users find the default game environment – the real dynamics – to be difficult and unintuitive, as indicated by their poor performance in the unassisted condition (top center and right plots) and their subjective evaluations (in Table 1). Our method observes suboptimal human play in the default environment, learns a setting of the game physics under which the observed human play would have been closer to optimal, then performs internal-to-real dynamics transfer to assist human users in achieving higher success rates and lower crash rates (top center and right plots). The learned internal dynamics has a slower game speed than the real dynamics (bottom left plot). The bottom center and right plots show successful (green) and failed (red) trajectories in the unassisted and assisted conditions.

that do not significantly deviate from those of an exact algorithm. Further results and experimental details are discussed in Sections 9.1.1 and 9.1.2 of the appendix.

**Learning rewards from misguided user demonstrations.** Standard IRL algorithms, such as Max-CausalEnt IRL [55], can fail to learn rewards from user demonstrations that are 'misguided', i.e., systematically suboptimal in the real world but near-optimal with respect to the user's internal dynamics. Our algorithm can learn the internal dynamics model, and we can then explicitly incorporate the learned internal dynamics into the MaxCausalEnt IRL algorithm to learn accurate rewards from misguided demonstrations. We assess this method on a simulated grid world navigation task. Figure 2 (right plot) supports our claim that standard IRL is ineffective at learning rewards from misguided user demonstrations. After using our algorithm to learn the internal dynamics and explicitly incorporating the learned internal dynamics into an IRL algorithm's model of the user, we see that it's possible to recover accurate rewards from these misguided demonstrations. Additional information on our experimental setup is available in Section 9.1.3 of the appendix.

In addition to comparing to the standard MaxCausalEnt IRL baseline, we also conducted a comparison (shown in Figure 2) with a variant of the Simultaneous Estimation of Rewards and Dynamics (SERD) algorithm [28] that simultaneously learns rewards and the internal dynamics instead of assuming that the internal dynamics are equivalent to the real dynamics. This baseline performs better than random, but still much worse than our method. This result is supported by the theoretical analysis in Armstrong et al. [2], which characterizes the difficulty of simultaneously deducing a human's rationality – in our case, their internal dynamics model – and their rewards from demonstrations.

### 5.2 User Study on the Lunar Lander Game

Our previous experiments were conducted with simulated expert behavior, which allowed us to control the corruption of the internal dynamics. However, it remains to be seen whether this model of suboptimality effectively reflects real human behavior. We test this hypothesis in the next experiment, which evaluates whether our method can learn the internal dynamics accurately enough to assist real users through internal-to-real dynamics transfer.

**Task description.** We use the Lunar Lander game from OpenAI Gym [9] (screenshot in Figure 1) to evaluate our algorithm with human users. The objective of the game is to land on the ground,

**Table 1:** Subjective evaluations of the Lunar Lander user study from 12 participants. Means reported below for responses on a 7-point Likert scale, where 1 = Strongly Disagree, 4 = Neither Disagree nor Agree, and 7 = Strongly Agree. $p$-values from a one-way repeated measures ANOVA with the presence of assistance as a factor influencing responses.

|  | $p$-value | Unassisted | Assisted |
| --- | --- | --- | --- |
| I enjoyed playing the game | $< .001$ | 3.92 | **5.92** |
| I improved over time | $< .0001$ | 3.08 | **5.83** |
| I didn't crash | $< .001$ | 1.17 | **3.00** |
| I didn't fly out of bounds | $< .05$ | 1.67 | **3.08** |
| I didn't run out of time | $> .05$ | 5.17 | 6.17 |
| I landed between the flags | $< .001$ | 1.92 | **4.00** |
| I understood how to complete the task | $< .05$ | 6.42 | **6.75** |
| I intuitively understood the physics of the game | $< .01$ | 4.58 | **6.00** |
| My actions were carried out | $> .05$ | 4.83 | 5.50 |
| My intended actions were carried out | $< .01$ | 2.75 | **5.25** |

without crashing or flying out of bounds, using two lateral thrusters and a main engine. The action space $\mathcal{A}$ consists of six discrete actions. The state $s \in \mathbb{R}^9$ encodes position, velocity, orientation, and the location of the landing site, which is one of nine values corresponding to $n = 9$ distinct tasks. The physics of the game are forward-simulated by a black-box function that takes as input seven hyperparameters, which include engine power and game speed. We manipulate whether or not the user receives internal-to-real dynamics transfer assistance using an internal dynamics model trained on their unassisted demonstrations. The dependent measures are the success and crash rates in each condition. The task and evaluation protocol are discussed further in Section 9.2 of the appendix.

**Analysis.** In the default environment, users appear to play as though they underestimate the strength of gravity, which causes them to crash into the ground frequently (see the supplementary videos). Figure 3 (bottom left plot) shows that our algorithm learns an internal dynamics model characterized by a slower game speed than the real dynamics, which makes sense since a slower game speed induces smaller forces and slower motion – conditions under which the users' action demonstrations would have been closer to optimal. These results support our claim that our algorithm can learn an internal dynamics model that explains user actions better than the real dynamics.

When unassisted, users often crash or fly out of bounds due to the unintuitive nature of the thruster controls and the relatively fast pace of the game. Figure 3 (top center and right plots) shows that users succeed significantly more often and crash significantly less often when assisted by internal-to-real dynamics transfer (see Section 9.2 of the appendix for hypothesis tests). The assistance makes the system feel easier to control (see the subjective evaluations in Table 1 of the appendix), less likely to tip over (see the supplementary videos), and move more slowly in response to user actions (assistance led to a 30% decrease in average speed). One of the key advantages of assistance was its positive effect on the rate at which users were able to switch between different actions: on average, unassisted users performed 18 actions per minute (APM), while assisted users performed 84 APM. Quickly switching between firing various thrusters enabled assisted users to better stabilize flight. These results demonstrate that the learned internal dynamics can be used to effectively assist the user through internal-to-real dynamics transfer, which in turn gives us confidence in the accuracy of the learned internal dynamics. After all, we cannot measure the accuracy of the learned internal dynamics by comparing it to the ground truth internal dynamics, which is unknown for human users.

## 6 Related Work

The closest prior work in intent inference and action understanding comes from inverse planning [3] and inverse reinforcement learning [37], which use observations of a user's actions to estimate the user's goal or reward function. We take a fundamentally different approach to intent inference: using action observations to estimate the user's beliefs about the world dynamics.

The simultaneous estimation of rewards and dynamics (SERD) instantiation of MaxCausalEnt IRL [28] aims to improve the sample efficiency of IRL by forcing the learned real dynamics model to explain observed state transitions as well as actions. The framework includes terms for the

demonstrator's beliefs of the dynamics, but the overall algorithm and experiments of Herman et al. [28] constrain those beliefs to be the same as the real dynamics. Our goal is to learn an internal dynamics model that may deviate from the real dynamics. To this end, we propose two new internal dynamics regularization techniques, multi-task training and the action intent prior (see Section 3.2), and demonstrate their utility for learning an internal dynamics model that differs from the real dynamics (see Section 5.1). We also conduct a user experiment that shows human actions in a game environment can be better explained by a learned internal dynamics model than by the real dynamics, and that augmenting user control with internal-to-real dynamics transfer results in improved game play. Furthermore, the SERD algorithm is well-suited to MDPs with a discrete state space, but intractable for continuous state spaces. Our method can be applied to MDPs with a continuous state space, as shown in Sections 5.1 and 5.2.

Golub et al. [22] propose an internal model estimation (IME) framework for brain-machine interface (BMI) control that learns an internal dynamics model from control demonstrations on tasks with linear-Gaussian dynamics and quadratic reward functions. Our work is (1) more general in that it places no restrictions on the functional form of the dynamics or the reward function, and (2) does not assume sensory feedback delay, which is the fundamental premise of using IME for BMI control.

Rafferty et al. [43, 41, 42] use an internal dynamics learning algorithm to infer a student's incorrect beliefs in online learning settings like educational games, and leverage the inferred beliefs to generate personalized hints and feedback. Our algorithm is more general in that it is capable of learning continuous parameters of the internal dynamics, whereas the cited work is only capable of identifying the internal dynamics given a discrete set of candidate models.

Modeling human error has a rich history in the behavioral sciences. Procrastination and other time-inconsistent human behaviors have been characterized as rational with respect to a cost model that discounts the cost of future action relative to that of immediate action [1, 32]. Systematic errors in human predictions about the future have been partially explained by cognitive biases like the availability heuristic and regression to the mean [50]. Imperfect intuitive physics judgments have been characterized as approximate probabilistic inferences made by a resource-bounded observer [26]. We take an orthogonal approach in which we assume that suboptimal behavior is primarily caused by incorrect beliefs of the dynamics, rather than uncertainty or biases in planning and judgment.

Humans are resource-bounded agents that must take into account the computational cost of their planning algorithm when selecting actions [24]. One way to trade-off the ability to find high-value actions for lower computational cost is to plan using a simplified, low-dimensional model of the dynamics [27, 19]. Evidence from the cognitive science literature suggests humans find it difficult to predict the motion of objects when multiple information dimensions are involved [40]. Thus, we arrive at an alternative explanation for why humans may behave near-optimally with respect to a dynamics model that differs from the real dynamics: even if users have perfect knowledge of the real dynamics, they may not have the computational resources to plan under the real dynamics, and instead choose to plan using a simplified model.

## 7 Discussion

**Limitations.** Although our algorithm models the soft $Q$ function with arbitrary neural network parameterizations, the internal dynamics parameterizations we use are smaller, with at most seven parameters for continuous tasks. Increasing the number of dynamics parameters would require a better approach to regularization than those proposed in Section 3.2.

**Summary.** We contribute an algorithm that learns a user's implicit beliefs about the dynamics of the environment from demonstrations of their suboptimal behavior in the real environment. Simulation experiments and a small-scale user study demonstrate the effectiveness of our method at recovering a dynamics model that explains human actions, as well as its utility for applications in shared autonomy and inverse reinforcement learning.

**Future work.** The ability to learn internal dynamics models from demonstrations opens the door to new directions of scientific inquiry, like estimating young children's intuitive theories of physics and psychology without eliciting verbal judgments [52, 18, 23]. It also enables applications that involve intent inference, including adaptive brain-computer interfaces for prosthetic limbs [12, 47] that help users perform control tasks that are difficult to fully specify.

# 8 Acknowledgements

We would like to thank Oleg Klimov for open-sourcing his implementation of the Lunar Lander game, which was originally developed by Atari in 1979, and inspired by the lunar modules built in the 1960s and 70s for the Apollo space program. We would also like to thank Eliezer Yudkowsky for the fanfiction novel, Harry Potter and the Methods of Rationality – Harry's misadventure with the rocket-assisted broomstick in chapter 59 inspired us to try to close the gap between intuitive physics and the real world. This work was supported in part by a Berkeley EECS Department Fellowship for first-year Ph.D. students, Berkeley DeepDrive, computational resource donations from Amazon, NSF IIS-1700696, and AFOSR FA9550-17-1-0308.

## Footnotes

[1]Our algorithm can in principle learn from demonstrations even when the rewards are unknown, but in practice we find that this relaxation usually makes learning the correct internal dynamics too difficult.

[2]We assume a discrete action space to simplify our exposition and experiments. Our algorithm can be extended to handle MDPs with a continuous action space using existing sampling methods [25].

[3]Another potential solution is sampling states to compute a Monte Carlo estimate of the integral.

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
