[Supplementary Material]

# 9 Appendix

This appendix contains additional discussion of experiments.

## 9.1 Experiments

### 9.1.1 Grid World Navigation

In Section 3.1 (in the main paper), we described two ways to adapt our algorithm to MDPs with a continuous state space: constraint sampling, and choosing a deterministic model of the internal dynamics. In this section, we evaluate our method on an MDP with a discrete state space in order to avoid the need for these two tricks. Our goal is to learn the internal dynamics and use it to assist the user through internal-to-real dynamics transfer. To sanity-check our algorithm and analyze its behavior under various hyperparameter settings and regularization choices, we implement a simple, deterministic grid world environment in which the simulated user attempts to navigate to a target position.

**Hypothesis.** Our algorithm is capable of learning accurate tabular representations of the internal dynamics for MDPs with a discrete state space. The two regularization schemes proposed in Section 3.2 (in the main paper) improve the quality of the learned internal dynamics model.

**Task description.** The state space consists of 49 states arranged in a 7x7 grid. The action space consists of four discrete actions that deterministically move the agent one step in each of the cardinal directions. The reward function emits a large bonus when the agent hits the target, a large penalty when the agent goes out of bounds, and includes a shaping term that rewards the agent for moving closer to the target. An episode lasts at most 100 timesteps. Each of the 49 states is a potential target, so the environment naturally yields 49 distinct tasks.

**Corrupting the internal dynamics.** To simulate suboptimal behavior, we create two users: one user whose action labels have been randomly scrambled in the same way at all states (e.g., the user's 'left' button actually moves them down instead, and this confusion is the same throughout the state space), and a different user whose action labels have been randomly scrambled in potentially different ways depending on which state they're in (e.g., 'left' takes them down in the top half of the grid, but takes them right in the bottom half). We refer to these two corruption models as 'globally scrambled actions' and 'locally scrambled actions' respectively. The users behave near-optimally with respect to their internal beliefs of the action labels, i.e., their internal dynamics, but because their beliefs about the action labels are incorrect, they act suboptimally in the actual environment.

**Evaluation.** We evaluate our method on its ability to learn the internal dynamics models of the simulated suboptimal users, i.e., on its ability to unscramble their actions, given demonstrations of their failed attempts to solve the task. The dependent measures are the next-state prediction accuracy of the learned internal dynamics compared to the ground truth internal dynamics, as well as the user's success rate when they are assisted with internal-to-real dynamics transfer (see Section 4.1 in the main paper) using the learned internal dynamics.

**Implementation details.** We use tabular representations of the $Q_{\theta_i}$ functions and the internal dynamics $T_\phi$. We collect 1000 demonstrations per training task, set $\rho = 2 \cdot 10^{-3}$ in Equation 6 (in the main paper), and enumerate the constraints in Equation 6 (in the main paper) instead of sampling.

**Manipulated factors.** We manipulate (1) whether or not the user receives assistance in the form of internal-to-real dynamics transfer using the learned internal dynamics model – a binary variable; (2) the number of training tasks on which we collect demonstrations – an integer-valued variable between 1 and 49; (3) the structure of the internal dynamics model – a categorical variable that can take on two values: state intent, which structures the internal dynamics in the usual way, or action intent, which uses Equation 7 (in the main paper) instead; and (4) the user's internal dynamics corruption scheme – a categorical variable that can take on two values: globally scrambled actions, or locally scrambled actions.

**Analysis.** Figure 4 provides overall support for the hypothesis that our method can effectively learn a tabular representation of the internal dynamics for an MDP with a discrete state space. The learned internal dynamics models are accurate with respect to the ground truth internal dynamics, especially when the user's internal dynamics corruption is systematic throughout the state space (top and bottom left plots).

**Figure 4:** Error bars show standard error on ten random seeds. Corrupting the internal dynamics of the simulated user by scrambling actions the same way at all states (top and bottom left plots) induces a much easier internal dynamics learning problem than scrambling actions differently at each state (top and bottom right plots).

We also compare the two regularization schemes discussed in Section 3.2 (in the main paper): training on multiple tasks, and imposing an action intent prior. Internal models are easier to learn when the user demonstrates their behavior on multiple training tasks, as shown by the increase in accuracy as the number of tasks (on the horizontal axis) increases. Regularizing the internal dynamics using action intent can be useful in some cases when the internal dynamics systematically deviate from the real dynamics, like when the user's actions are scrambled in the same way throughout the state space (top and bottom left plots, compare orange vs. teal curve), but can have a varying effect in other cases where the internal dynamics are severely biased away from reality, like when the action scrambling varies between states (top and bottom right plots, compare orange vs. teal curve).

### 9.1.2 2D Continuous Navigation

In the previous section, we adopted a tabular grid world environment in order to avoid constraint sampling in Equation 6 (in the main paper). Now, we would like to show that our method still works even when we sample constraints to be able to handle a continuous state space.

**Hypothesis.** Our algorithm can learn accurate continuous representations of the internal dynamics for MDPs with a continuous state space.

**Task description.** As mentioned in Section 1 (in the main paper), a classic study in cognitive science shows that people's intuitive judgments about the physics of projectile motion are closer to Aristotelian impetus theory than to true Newtonian dynamics [11]. In other words, people tend to ignore or underestimate the effects of inertia. Inspired by this study, we create a simple 2D environment in a which a simulated user must move a point mass from its initial position to a target position as quickly as possible using a discrete action space of four arrow keys and continuous, low-dimensional observations of position and velocity. The system follows deterministic, linear dynamics. Formally,

$$\boldsymbol{x_{t+1}} = A\boldsymbol{x_t} + B\boldsymbol{u_t} \tag{9}$$

where $\boldsymbol{x} = (x, y, v_x, v_y)^{\mathsf{T}}$ denotes the state, $\boldsymbol{u} \in \{(\pm 0.01, 0)^{\mathsf{T}}, (0, \pm 0.01)^{\mathsf{T}}\}$ denotes the control,

$$A = \begin{pmatrix} 1 & 0 & a_{13} & 0 \\ 0 & 1 & 0 & a_{24} \\ 0 & 0 & a_{33} & 0 \\ 0 & 0 & 0 & a_{44} \end{pmatrix}, B = \begin{pmatrix} b_{11} & 0 \\ 0 & b_{22} \\ b_{31} & 0 \\ 0 & b_{42} \end{pmatrix}.$$

At the beginning of each episode, the state is reset to $\boldsymbol{x_0} = (x_0 \sim \mathrm{Unif}(0, 1), y_0 \sim \mathrm{Unif}(0, 1), 0, 0)$. The episode ends if the agent reaches the target (gets within a 0.02 radius around the target), goes out of bounds (outside the unit square), or runs out of time (takes longer than 200 timesteps).

**Figure 5:** Our method is able to assist the simulated suboptimal user through internal-to-real dynamics transfer. Sample paths followed by the unassisted and assisted user on a single task are shown above. Red paths end out of bounds; green, at the target marked by a yellow star.

**Corrupting the internal dynamics.** In the simulation, actions control acceleration and inertia exists; in other words, $b_{11} = b_{22} = 0$ and the rest of the parameters are set to 1. We create a simulated suboptimal user that behaves as if their actions control velocity and inertia does not exist, which causes them to follow trajectories that oscillate around the target or go out of bounds. The user behaves near-optimally with respect to their internal beliefs about the dynamics, but because their beliefs are incorrect, they act suboptimally in the real environment.

**Evaluation.** As before, we evaluate our method on its ability to learn the internal dynamics models of the simulated suboptimal user given demonstrations of their failed attempts to solve the task. We manipulate whether or not the user receives assistance in the form of internal-to-real dynamics transfer using the learned internal dynamics model. The dependent measures are the L2 error of the learned internal dynamics model parameters with respect to the ground truth internal dynamics parameters, and the success and crash rates of the user in each condition.

**Implementation details.** We fix the number of training tasks at $n = 49$, and use a multi-layer perceptron with one hidden layer of 32 units to represent the $Q_{\boldsymbol{\theta}_i}$ functions. We use a linear model based on Equation 9 to represent the internal dynamics $T_\phi$, in which $\hat{a}_{13}, \hat{a}_{24}, \hat{a}_{33}, \hat{a}_{44}, \hat{b}_{11}, \hat{b}_{22}, \hat{b}_{31}, \hat{b}_{42} \in [0, 1]$. We collect 1000 demonstrations per training task, set $\rho = 2$ in Equation 6 (in the main paper), and sample constraints in Equation 6 (in the main paper) by collecting 500 rollouts of a random policy in the real world (see Section 3.1 in the main paper for details).

**Analysis.** Our algorithm correctly learns the following internal dynamics parameters: (1) $\hat{a}_{33} = \hat{a}_{44} = 0$ in the learned internal dynamics, which corresponds to the user's belief that inertia does not exist; (2) $\hat{b}_{11} = \hat{b}_{22} = 1$ and $\hat{b}_{31} = \hat{b}_{42} = 0$ in the learned internal dynamics, which matches the user's belief that they have velocity control instead of acceleration control. The learned internal dynamics maintains $\hat{a}_{13} = \hat{a}_{24} = 1$, as in the real dynamics, which makes sense since the user's behavior is consistent with these parameters. Figure 5 (left plot) demonstrates the stability of our algorithm in converging to the correct internal dynamics.

Figure 5 (center and right plots) shows examples of trajectories followed by the simulated suboptimal user on their own and when they are assisted by internal-to-real dynamics transfer. The assisted user tends to move directly to the target instead of oscillating around it or missing it altogether.

### 9.1.3 Learning Rewards from Misguided User Demonstrations

The previous simulation experiments show that our algorithm can learn internal dynamics models that are useful for shared autonomy. Now, we explore a different application of our algorithm: learning rewards from demonstrations generated by a user with a misspecified internal dynamics model. In order to compare to prior methods that operate on tabular MDPs, we adopt the grid world setup from Section 9.1.1, with globally scrambled actions as the internal dynamics corruption scheme.

**Hypothesis.** Standard IRL algorithms can fail to learn rewards from user demonstrations that are 'misguided', i.e., suboptimal in the real world but near-optimal with respect to the user's internal dynamics. Our algorithm can learn the internal dynamics model, then we can explicitly incorporate the learned internal dynamics into standard IRL to learn accurate rewards from misguided demonstrations.

**Evaluation.** We evaluate our method on its ability to learn an internal dynamics model that is useful for 'debiasing' misguided user demonstrations, which serve as input to the MaxCausalEnt

IRL algorithm described in Section 4.2 (in the main paper). We manipulate whether we use the learned internal dynamics, or assume the internal dynamics to be the same as the real dynamics. The dependent measure is the true reward collected by a policy that is optimized for the rewards learned by MaxCausalEnt IRL.

**Implementation details.** We implement the MaxCausalEnt IRL algorithm [55, 28]. The reward function is represented as a table $R(s)$.

**Analysis.** Figure 2 (in the main paper, right plot) supports our claim that standard IRL is not capable of learning rewards from misguided user demonstrations, and that after using our algorithm to learn the internal dynamics and explicitly incorporating the learned internal dynamics into an IRL algorithm's behavioral model of the user, we learn accurate rewards.

## 9.2  User Study on the Lunar Lander Game

**Task description.** The reward function emits a large bonus at the end of the episode for landing between the flags, a large penalty for crashing or going out of bounds, and is shaped to penalize speed, tilt, and moving away from the landing site. The physics of the game are deterministic.

**Evaluation protocol.** We evaluate our method on its ability to learn the internal dynamics models of human users given demonstrations of their failed attempts to solve the task in the default environment. We manipulate whether or not the user receives assistance in the form of internal-to-real dynamics transfer using the learned internal dynamics. The dependent measures are the success and crash rates in each condition.

**Implementation details.** We fix the number of training tasks at $n = 9$ and use a multi-layer perceptron with one hidden layer of 32 units to represent the $Q_{\theta_i}$ functions. We collect 5 demonstrations per training task per user, set $\rho = 2 \cdot 10^{-3}$ in Equation 6 (in the main paper), and sample constraints in Equation 6 (in the main paper) by collecting 100 rollouts of a random policy in the real world (see Section 3.1 in the main paper for details).

The physics of the game are governed in part by a configurable vector $\psi \in \mathbb{R}^7$ that encodes engine power, game speed, and other relevant parameters. Since we cannot readily access an analytical expression of the dynamics, only a black-box function that forward-simulates the dynamics, we cannot simply parameterize our internal dynamics model using $\psi$ (see Section 3.1 in the main paper for details). Instead, we draw 100 random samples of $\psi$ and represent our internal dynamics model as a categorical probability distribution over the samples. In other words, we approximate the continuous space of possible internal dynamics models using a discrete set of samples. To accommodate this representation, we modify Equation 4 (in the main paper):

$$\delta_{\theta_i,\phi}(s,a) \triangleq Q_{\theta_i}(s,a) - \mathbb{E}_{j \sim \mathrm{Cat}(100,\phi)} \left[ \int_{s' \in \mathcal{S}} T_{\psi_j}(s'|s,a) \left( R_i(s,a,s') + \gamma V_{\theta_i}(s') \right) \mathrm{d}s' \right]$$

$$= Q_{\theta_i}(s,a) - \sum_{j=1}^{100} \phi_j \cdot \int_{s' \in \mathcal{S}} T_{\psi_j}(s'|s,a) \left( R_i(s,a,s') + \gamma V_{\theta_i}(s') \right) \mathrm{d}s'.$$

**Subject allocation.** We recruited 9 male and 3 female participants, with an average age of 24. Each participant was provided with the rules of the game and a short practice period of 9-18 episodes to familiarize themselves with the controls and dynamics. Each user played in both conditions: unassisted, and assisted. To avoid the confounding effect of humans learning to play the game better over time, we counterbalanced the order of the two conditions. Each condition lasted 45 episodes.

Counterbalancing the order of the two conditions sometimes requires testing the user in the assisted condition *before* the unassisted condition, which begs the question: where do the demonstrations used to train the internal dynamics model used in internal-to-real dynamics transfer assistance come from, if not the data from the unassisted condition? We train the internal dynamics model used to assist the $k$-th participant on the pooled, unassisted demonstrations of all previous participants $\{1, 2, ..., k-1\}$. After the $k$-th participant completes both conditions, we train an internal dynamics model solely on unassisted demonstrations from the $k$-th participant and verify that the resulting internal dynamics model is the same as the one used to assist the $k$-th participant.

**Analysis.** After inspecting the results of our random search over the internal dynamics space, we found that the game speed parameter in $\psi$ had a much larger influence on the quality of the learned

**Figure 6:** Assistance in the form of internal-to-real dynamics transfer increases success rates and decreases crash rates.

internal dynamics and the resulting internal-to-real dynamics transfer than the other six parameters. Hence, in Figure 3 (in the main paper, bottom left plot), we show the results of a grid search on the game speed parameter, holding the other six parameters constant at their default values. The game speed parameter governs the size of the time delta with which the game engine advances the physics simulation at each discrete step. This parameter indirectly controls the strength of the forces in the game physics: smaller time deltas lead to smaller forces and generally slower motion, and larger deltas to larger forces and consequently faster motion.

We ran a one-way repeated measures ANOVA with the presence of assistance as a factor influencing success and crash rates, and found that $f(1, 11) = 109.58, p < 0.0001$ for the success rate and $f(1, 11) = 126.33, p < 0.0001$ for the crash rate. The assisted user succeeds significantly more often and crashes significantly less often than the unassisted user. Figure 6 shows the raw data.