[Reviews · NeurIPS 2018]

Reviewer 1



The paper investigates the problem of inferring an agent's belief of the system dynamics of an MDP, given demonstrations of its behavior and the reward function it was optimizing. Knowledge of this internal belief can be used for Inverse Reinforcement Learning of an unknown task in the same environment. Furthermore, given the action provided by the agent, its intended action on the true dynamics can be inferred. This allows for assistive tele-operation, by applying the intended actions to the system instead of the provided ones. The proposed method models the agent using the model derived in maximum causal entropy inverse reinforcement learning. The internal belief of the system dynamics is inferred by maximizing the likelihood of the demonstrations with respect to the parameters of the dynamic model. Human studies on the lunar lander Atari game demonstrate that such assistive systems based on inferred internal models can help the user to solve the given task. Related problems in an MDP are RL, which learns behavior from dynamics and reward, as well as IRL, which learns reward from dynamics and behavior. This work tackles the problem of learning dynamics from behavior and reward which has gotten little attention from the community. There has been some prior work on this problem that was also mentioned in the paper. Herman et al. (2016) considered the problem of simultaneously inferring dynamics and reward function from expert demonstrations while also building on MaxCausalEnt-IRL. Although, the work by Herman et al. allows for internal models that differ from the true dynamics, they focused on inferring the true dynamics and assumed in the experiments that the internal model was correct. Golub et al. (2013) infer an internal model from reward function and dynamics. They derived an EM-algorithm that they applied for learning the internal model of Rhesus monkeys that were using a brain computer interface. However they consider a slightly different scenario where the subject's internal estimate of the state differs from the true state due to sensory feedback delay. The paper under review focuses on the internal model (and assumes that the true dynamics are known for the action intent prior) and in contrast to previous work puts more focus on the use cases. Using an internal model to correct the user's actions is novel to my knowledge and could be useful for real applications like tele-operating a robot. Also the application to IRL is interesting, however I think this scenario is less relevant, because we assume that 1) we want to infer the reward function of an agent that 2) is willing to perform demonstrations on different tasks and we 3) assume that its internal model is significantly different from the true dynamics. The proposed method is based on a sound optimization problem by maximizing the likelihood of the demonstrations assuming the expert model of MaxCausalEnt-IRL. Yet, I think it would be more principled to derive both the expert model as well as the objective from a single optimization problem (e.g. similar to MaxEnt Reinforcement Learning but optimizing dynamics rather than policy). This might also avoid the need for the penalty method, which is a simple but rather inefficient and unstable approach. The proposed method does in general tend to prefer the easy route, e.g. by assuming deterministic dynamics (although humans seem to take uncertainty into account [1]) and by using uniform sampling in section 3.1. which should lead to rather high variance. Nevertheless, the method is still sound and, in my opinion, the method itself is not the main contribution of the paper anyway. Indeed, it is more important to raise the awareness and show the usefulness of such little explored research branches, which is I think the main contribution of the proposed paper. The paper is well written and easy to follow. However, I think it would be better if the experiments would be better separated, e.g. by removing everything relating to the lunar lander game from section 5.1 and calling the section something like "Gridworld experiments". The gridworld experiments should be briefly described also in the paper itself. Questions to the authors: For learning the internal model, we need the user to optimize a given reward function. For some problems, specifying a reward is easy (e.g. Atari games). However, how would you make a human optimize a reward according to maxent-model for a real application? Concerning the action intend prior: Wouldn't it be simpler to 1) solve the MDP using the true dynamics and 2) compute the mapping (f_theta) from actions to intended actions as the maximum likelihood conditional distribution? How are the demonstrations on the gridworld collected? If whole trajectories were collected at once, I assume that the user knew how the dynamics are scrambled since the proposed method can not handle changing beliefs. However, if the user knew the true dynamics, why would his internal belief differ? Wouldn't the sub-optimality of the user's behavior be caused solely by its inability to choose the correct actions under real-time constraints? Or were the trajectories collected by independently querying the users action for a given state without revealing the consequences of this action? [1] Belousov, Boris, et al. "Catching heuristics are optimal control policies." Advances in Neural Information Processing Systems. 2016. Edit: I read the the other rebuttal / reviews and did not see a reason to change my assessment.

Reviewer 2



This paper proposes a scheme that learns the internal dynamics of an angent, under the assumption that the agent's actions are nearly optimal under its internal dynamics (which deviate from the real ones). The paper discusses some uses for learning such an internal model and performs tests to validate the assumption that IRL for humans is aided by the proposed scheme, and the underlying idea (that humans move optimally wrt a wrong model). While there are definetely some issues, which will be discussed below, I commend the authors for this work. I found the paper a joy to read, it seems free of technical errors and makes a clear contribution to the field. ========== QUALITY ============== - A large amount of the paper's abstract/introduction discusses the IRL setting, whereas most of the experiments (including the human experiment) concerns the shared autonomy setting. This should be better balanced. - Ideally, an extra experiment on the lunar lander, with slightly different objective (say hovering) would make the story-line in the paper complete. While it is difficult to assess the improvement of the IRL for human tests in general, a simple task such as hovering would provide some insights. ========== CLARITY ============== - The problem of regularizing the internal dynamics model is quite important, and the fact that it exists (due to the way the model-learning is setup) should be announced clearly. I suggest adding a sentence or two in both the introduction and the start of Section 3. - The whole choice of assuming optimality with a possibly completely incorrect model has many implication. The proposed regularization reduces the potential 'incorrectness' of the model somewhat, but it would be interesting to explore a more principled way of combining model error and non-optimality. The limitations of the current approach in this respect should be made clearer in the paper. - In equation 4 and 5, the match between the delta is suggested to be a constraint. In effect, you solve this with a penalty function, essentially turning delta^2 into an additional objective. The two terms in equation 6 are better explained as two cost functions, skipping the 'constraint'-explanation. That way the rather optimistic assumption that delta can be set to zero is also removed. ========== ORIGINALITY/SIGNIFICANCE ========= - The paper takes an interesting idea, and does a good job of exploring the solution. - The proposed algorithm is promising, but will not have an immediate large impact on IRL and shared autonomy. However, the approach taken might have such an impact in the future.

Reviewer 3



In this paper, the authors study the inverse reinforcement learning from a new perspective, i.e., they try to model the human intended dynamics (which is referred to as the 'internal dynamics'), also they show how the learned internal dynamics can be applied to the 'simulation-to-real transfer' scheme. Several examples are provided to support their assumption (i.e., the human recognized dynamics might be different from the real dynamics) and main results. One problem I am mainly concerned is that the authors first assume the internal dynamics severely deviates from the real dynamics, and then argue that it is not possible to learn the correct reward function using traditional IRL. Intuitively, if the user has very bad interpretation of the dynamics, the user will usually adapt their rewards accordingly, namely the dynamics (i.e., transition probability) and the reward function should be learned together. In the evaluations, the authors explained that the classical IRL can not perform well and meanwhile compare the classical IRL with their work, which looks kinda unfair, after all the assumptions behind both approaches are quite different. It would be more convincing if the authors could learn the human intention dynamics and the reward function simultaneously, and then compare this hybrid learning with their work. Other small issues that arise when I go through this paper are provided as follows: - in line 36 it is not clear about the difference 'suboptimal' and the 'near-optimal', in my opinion the second 'near-optimal' should be 'optimal' since the user should naturally choose the best action based on his recognition and understanding. - in line 97, the authors metion the 'near-optimal' again, but in line 197, it becomes 'but optimal with respect.....' Since this description is very important, the authors should check them carefully. - in line 133, it is unclear how to choose this 'random policy' as this will influence the sampling significantly, also this information is missing in the evaluations. - for equ.(7), I found that it is not very straightforward to interpret the function 'f', it seems this function models the action error between desired action and real action, please clarify the notations of 'a' and 'a^{int}'. - in line 186, it would be more clear if the authors mention fig.1 after equ. (8) - in fig. 2, the pure IRL performs poorly, so what happened if the internal dynamics is slightly different from the real dynamics?